# Concrete Dropout

**Yarin Gal**
yarin.gal@eng.cam.ac.uk
University of Cambridge
and Alan Turing Institute, London

**Jiri Hron**
jh2084@cam.ac.uk
University of Cambridge

**Alex Kendall**
agk34@cam.ac.uk
University of Cambridge

## Abstract

Dropout is used as a practical tool to obtain uncertainty estimates in large vision models and reinforcement learning (RL) tasks. But to obtain well-calibrated uncertainty estimates, a grid-search over the dropout probabilities is necessary—a prohibitive operation with large models, and an impossible one with RL. We propose a new dropout variant which gives improved performance and better calibrated uncertainties. Relying on recent developments in Bayesian deep learning, we use a continuous relaxation of dropout's discrete masks. Together with a principled optimisation objective, this allows for automatic tuning of the dropout probability in large models, and as a result faster experimentation cycles. In RL this allows the agent to adapt its uncertainty dynamically as more data is observed. We analyse the proposed variant extensively on a range of tasks, and give insights into common practice in the field where larger dropout probabilities are often used in deeper model layers.

## 1 Introduction

Well-calibrated uncertainty is crucial for many tasks in deep learning. From the detection of adversarial examples [25], through an agent exploring its environment safely [10, 18], to analysing failure cases in autonomous driving vision systems [20]. Tasks such as these depend on good uncertainty estimates to perform well, with miscalibrated uncertainties in reinforcement learning (RL) having the potential to lead to over-exploration of the environment. Or, much worse, miscalibrated uncertainties in an autonomous driving vision systems leading to its failure to detect its own ignorance about the world, resulting in the loss of human life [29].

A principled technique to obtaining uncertainty in models such as the above is *Bayesian inference*, with *dropout* [9, 14] being a practical inference approximation. In dropout inference the neural network is trained with dropout at training time, and at test time the output is evaluated by dropping units randomly to generate samples from the predictive distribution [9]. But to get well-calibrated uncertainty estimates it is necessary to adapt the dropout probability as a variational parameter to the data at hand [7]. In previous works this was done through a *grid-search* over the dropout probabilities [9]. Grid-search can pose difficulties though in certain tasks. Grid-search is a prohibitive operation with large models such as the ones used in Computer Vision [19, 20], where multiple GPUs would be used to train a single model. Grid-searching over the dropout probability in such models would require either an immense waste of computational resources, or extremely prolonged experimentation cycles. More so, the number of possible per-layer dropout configurations grows exponentially as the number of model layers increases. Researchers have therefore restricted the grid-search to a small number of possible dropout values to make such search feasible [8], which in turn might hurt uncertainty calibration in vision models for autonomous systems.

In other tasks a grid-search over the dropout probabilities is impossible altogether. In tasks where the amount of data changes over time, for example, the dropout probability should be decreased as the amount of data increases [7]. This is because the dropout probability has to diminish to zero in the limit of data—with the model explaining away its uncertainty completely (this is explained in more detail in §2). RL is an example setting where the dropout probability has to be adapted dynamically.

The amount of data collected by the agent increases steadily with each episode, and in order to reduce the agent's uncertainty, the dropout probability must be decreased. Grid-searching over the dropout probability is impossible in this setting, as the agent will have to be reset and re-trained with the entire data with each new acquired episode. A method to tune the dropout probability which results in good accuracy *and uncertainty estimates* is needed then.

Existing literature on tuning the dropout probability is sparse. Current methods include the optimisation of $\alpha$ in Gaussian dropout following its variational interpretation [23], and overlaying a binary belief network to optimise the dropout probabilities as a function of the inputs [2]. The latter approach is of limited practicality with large models due to the increase in model size. With the former approach [23], practical use reveals some unforeseen difficulties [28]. Most notably, the $\alpha$ values have to be truncated at $1$, as the KL approximation would diverge otherwise. In practice the method under-performs.

In this work we propose a new practical dropout variant which can be seen as a continuous relaxation of the discrete dropout technique. Relying on recent techniques in Bayesian deep learning [16, 27], together with appropriate regularisation terms derived from dropout's Bayesian interpretation, our variant allows the dropout probability to be tuned using gradient methods. This results in better-calibrated uncertainty estimates in large models, avoiding the coarse and expensive grid-search over the dropout probabilities. Further, this allows us to use dropout in RL tasks in a principled way.

We analyse the behaviour of our proposed dropout variant on a wide variety of tasks. We study its ability to capture different types of uncertainty on a simple synthetic dataset with known ground truth uncertainty, and show how its behaviour changes with increasing amounts of data versus model size. We show improved accuracy and uncertainty on popular datasets in the field, and further demonstrate our variant on large models used in the Computer Vision community, showing a significant reduction in experiment time as well as improved model performance and uncertainty calibration. We demonstrate our dropout variant in a model-based RL task, showing that the agent automatically reduces its uncertainty as the amount of data increases, and give insights into common practice in the field where a small dropout probability is often used with the shallow layers of a model, and a large dropout probability used with the deeper layers.

## 2 Background

In order to understand the relation between a model's uncertainty and the dropout probability, we start with a slightly philosophical discussion of the different types of uncertainty available to us. This discussion will be grounded in the development of new tools to better understand these uncertainties in the next section.

Three types of uncertainty are often encountered in Bayesian modelling. Epistemic uncertainty captures our ignorance about the models most suitable to explain our data; Aleatoric uncertainty captures noise inherent in the environment; Lastly, predictive uncertainty conveys the model's uncertainty in its output. Epistemic uncertainty reduces as the amount of observed data increases—hence its alternative name "reducible uncertainty". When dealing with models over functions, this uncertainty can be captured through the range of possible functions and the probability given to each function. This uncertainty is often summarised by generating function realisations from our distribution and estimating the variance of the functions when evaluated on a fixed set of inputs. Aleatoric uncertainty captures noise sources such as measurement noise—noises which cannot be explained away even if more data were available (although this uncertainty *can* be reduced through the use of higher precision sensors for example). This uncertainty is often modelled as part of the likelihood, at the top of the model, where we place some noise corruption process on the function's output. Gaussian corrupting noise is often assumed in regression, although other noise sources are popular as well such as Laplace noise. By inferring the Gaussian likelihood's precision parameter $\tau$ for example we can estimate the amount of aleatoric noise inherent in the data.

Combining both types of uncertainty gives us the predictive uncertainty—the model's confidence in its prediction, taking into account noise it can explain away and noise it cannot. This uncertainty is often obtained by generating multiple functions from our model and corrupting them with noise (with precision $\tau$). Calculating the variance of these outputs on a fixed set of inputs we obtain the model's predictive uncertainty. This uncertainty has different properties for different inputs. Inputs near the training data will have a smaller epistemic uncertainty component, while inputs far away

from the training data will have higher epistemic uncertainty. Similarly, some parts of the input space might have larger aleatoric uncertainty than others, with these inputs producing larger measurement error for example. These different types of uncertainty are of great importance in fields such as AI safety [1] and autonomous decision making, where the model's epistemic uncertainty can be used to avoid making uninformed decisions with potentially life-threatening implications [20].

When using dropout neural networks (or any other stochastic regularisation technique), a randomly drawn masked weight matrix corresponds to a function draw [7]. Therefore, the dropout probability, together with the weight configuration of the network, determine the magnitude of the epistemic uncertainty. For a fixed dropout probability $p$, high magnitude weights will result in higher output variance, i.e. higher epistemic uncertainty. With a fixed $p$, a model wanting to decrease its epistemic uncertainty will have to reduce its weight magnitude (and set the weights to be exactly zero to have zero epistemic uncertainty). Of course, this is impossible, as the model will not be able to explain the data well with zero weight matrices, therefore some balance between desired output variance and weight magnitude is achieved[1]. For uncertainty representation, this can be seen as a degeneracy with the model when the dropout probability is held fixed.

Allowing the probability to change (for example by grid-searching it to maximise validation log-likelihood [9]) will let the model decrease its epistemic uncertainty by choosing smaller dropout probabilities. But if we wish to replace the grid-search with a gradient method, we need to define an optimisation objective to optimise $p$ with respect to. This is not a trivial thing, as our aim is not to maximise model performance, but rather to obtain *good epistemic uncertainty*. What is a suitable objective for this? This is discussed next.

## 3   Concrete Dropout

One of the difficulties with the approach above is that grid-searching over the dropout probability can be expensive and time consuming, especially when done with large models. Even worse, when operating in a continuous learning setting such as reinforcement learning, the model should collapse its epistemic uncertainty as it collects more data. When grid-searching this means that the data has to be set-aside such that a new model could be trained with a smaller dropout probability when the dataset is large enough. This is infeasible in many RL tasks. Instead, the dropout probability can be optimised using a gradient method, where we seek to minimise some objective with respect to (w.r.t.) that parameter.

A suitable objective follows dropout's variational interpretation [7]. Following the variational interpretation, dropout is seen as an approximating distribution $q_\theta(\boldsymbol{\omega})$ to the posterior in a Bayesian neural network with a set of random weight matrices $\boldsymbol{\omega} = \{\mathbf{W}_l\}_{l=1}^L$ with $L$ layers and $\theta$ the set of variational parameters. The optimisation objective that follows from the variational interpretation can be written as:

$$\widehat{\mathcal{L}}_{\text{MC}}(\theta) = -\frac{1}{M}\sum_{i \in S}\log p(\mathbf{y}_i|\mathbf{f}^{\boldsymbol{\omega}}(\mathbf{x}_i)) + \frac{1}{N}\text{KL}(q_\theta(\boldsymbol{\omega})||p(\boldsymbol{\omega})) \tag{1}$$

with $\theta$ parameters to optimise, $N$ the number of data points, $S$ a random set of $M$ data points, $\mathbf{f}^{\boldsymbol{\omega}}(\mathbf{x}_i)$ the neural network's output on input $\mathbf{x}_i$ when evaluated with weight matrices realisation $\boldsymbol{\omega}$, and $p(\mathbf{y}_i|\mathbf{f}^{\boldsymbol{\omega}}(\mathbf{x}_i))$ the model's likelihood, e.g. a Gaussian with mean $\mathbf{f}^{\boldsymbol{\omega}}(\mathbf{x}_i)$. The KL term $\text{KL}(q_\theta(\boldsymbol{\omega})||p(\boldsymbol{\omega}))$ is a "regularisation" term which ensures that the approximate posterior $q_\theta(\boldsymbol{\omega})$ does not deviate too far from the prior distribution $p(\boldsymbol{\omega})$. A note on our choice for a prior is given in appendix B. Assume that the set of variational parameters for the dropout distribution satisfies $\theta = \{\mathbf{M}_l, p_l\}_{l=1}^L$, a set of mean weight matrices and dropout probabilities such that $q_\theta(\boldsymbol{\omega}) = \prod_l q_{\mathbf{M}_l}(\mathbf{W}_l)$ and $q_{\mathbf{M}_l}(\mathbf{W}_l) = \mathbf{M}_l \cdot \text{diag}[\text{Bernoulli}(1-p_l)^{K_l}]$ for a single random weight matrix $\mathbf{W}_l$ of dimensions $K_{l+1}$ by $K_l$. The KL term can be approximated well following [7]

$$\text{KL}(q_\theta(\boldsymbol{\omega})||p(\boldsymbol{\omega})) = \sum_{l=1}^L \text{KL}(q_{\mathbf{M}_l}(\mathbf{W}_l)||p(\mathbf{W}_l)) \tag{2}$$

$$\text{KL}(q_{\mathbf{M}}(\mathbf{W})||p(\mathbf{W})) \propto \frac{l^2(1-p)}{2}||\mathbf{M}||^2 - K\mathcal{H}(p) \tag{3}$$

with

$$\mathcal{H}(p) := -p \log p - (1-p) \log(1-p) \tag{4}$$

the *entropy* of a Bernoulli random variable with probability $p$.

The entropy term can be seen as a *dropout regularisation* term. This regularisation term depends on the dropout probability $p$ alone, which means that the term is constant w.r.t. model weights. For this reason the term can be omitted when the dropout probability is not optimised, but the term is crucial when it *is* optimised. Minimising the KL divergence between $q_{\mathbf{M}}(\mathbf{W})$ and the prior is equivalent to maximising the entropy of a Bernoulli random variable with probability $1 - p$. This pushes the dropout probability towards $0.5$—the highest it can attain. The scaling of the regularisation term means that large models will push the dropout probability towards $0.5$ much more than smaller models, but as the amount of data $N$ increases the dropout probability will be pushed towards $0$ (because of the first term in eq. (1)).

We need to evaluate the derivative of the last optimisation objective eq. (1) w.r.t. the parameter $p$. Several estimators are available for us to do this: for example the score function estimator (also known as a likelihood ratio estimator and Reinforce [6, 12, 30, 35]), or the pathwise derivative estimator (this estimator is also referred to in the literature as the re-parametrisation trick, infinitesimal perturbation analysis, and stochastic backpropagation [11, 22, 31, 34]). The score function estimator is known to have extremely high variance in practice, making optimisation difficult. Following early experimentation with the score function estimator, it was evident that the increase in variance was not manageable. The pathwise derivative estimator is known to have much lower variance than the score function estimator in many applications, and indeed was used by [23] with Gaussian dropout. However, unlike the Gaussian dropout setting, in our case we need to optimise the parameter of a Bernoulli distributions. The pathwise derivative estimator assumes that the distribution at hand can be re-parametrised in the form $g(\theta, \epsilon)$ with $\theta$ the distribution's parameters, and $\epsilon$ a random variable which does not depend on $\theta$. This cannot be done with the Bernoulli distribution.

Instead, we replace dropout's discrete Bernoulli distribution with its continuous relaxation. More specifically, we use the Concrete distribution relaxation. This relaxation allows us to re-parametrise the distribution and use the low variance pathwise derivative estimator instead of the score function estimator.

The Concrete distribution is a continuous distribution used to approximate discrete random variables, suggested in the context of latent random variables in deep generative models [16, 27]. One way to view the distribution is as a relaxation of the "max" function in the Gumbel-max trick to a "softmax" function, which allows the discrete random variable $\mathbf{z}$ to be written in the form $\tilde{\mathbf{z}} = g(\theta, \epsilon)$ with parameters $\theta$, and $\epsilon$ a random variable which does not depend on $\theta$.

We will concentrate on the binary random variable case (i.e. a Bernoulli distribution). Instead of sampling the random variable from the discrete Bernoulli distribution (generating zeros and ones) we sample realisations from the *Concrete distribution* with some temperature $t$ which results in values in the interval $[0, 1]$. This distribution concentrates most mass on the boundaries of the interval $0$ and $1$. In fact, for the one dimensional case here with the Bernoulli distribution, the Concrete distribution relaxation $\tilde{\mathbf{z}}$ of the Bernoulli random variable $\mathbf{z}$ reduces to a simple sigmoid distribution which has a convenient parametrisation:

$$\tilde{\mathbf{z}} = \text{sigmoid}\left(\frac{1}{t} \cdot \big(\log p - \log(1-p) + \log u - \log(1-u)\big)\right) \tag{5}$$

with uniform $u \sim \text{Unif}(0, 1)$. This relation between $u$ and $\tilde{\mathbf{z}}$ is depicted in figure 10 in appendix A. Here $u$ is a random variable which does not depend on our parameter $p$. The functional relation between $\tilde{\mathbf{z}}$ and $u$ is differentiable w.r.t. $p$.

With the Concrete relaxation of the dropout masks, it is now possible to optimise the dropout probability using the pathwise derivative estimator. We refer to this Concrete relaxation of the dropout masks as *Concrete Dropout*. A Python code snippet for Concrete dropout in Keras [5] is given in appendix C, spanning about 20 lines of code, and experiment code is given online[2]. We next assess the proposed dropout variant empirically on a large array of tasks.

# 4 Experiments

We next analyse the behaviour of our proposed dropout variant on a wide variety of tasks. We study how our dropout variant captures different types of uncertainty on a simple synthetic dataset with known ground truth uncertainty, and show how its behaviour changes with increasing amounts of data versus model size (§4.1). We show that Concrete dropout matches the performance of hand-tuned dropout on the UCI datasets (§4.2) and MNIST (§4.3), and further demonstrate our variant on large models used in the Computer Vision community (§4.4). We show a significant reduction in experiment time as well as improved model performance and uncertainty calibration. Lastly, we demonstrate our dropout variant in a model-based RL task extending on [10], showing that the agent correctly reduces its uncertainty dynamically as the amount of data increases (§4.5).

We compare the performance of hand-tuned dropout to our Concrete dropout variant in the following experiments. We chose not to compare to Gaussian dropout in our experiments, as when optimising Gaussian dropout's $\alpha$ following its variational interpretation [23], the method is known to under-perform [28] (however, Gal [7] compared Gaussian dropout to Bernoulli dropout and found that when optimising the dropout probability by hand, the two methods perform similarly).

## 4.1 Synthetic data

The tools above allow us to separate both epistemic and aleatoric uncertainties with ease. We start with an analysis of how different uncertainties behave with different data sizes. For this we optimise both the dropout probability $p$ as well as the (per point) model precision $\tau$ (following [20] for the latter one). We generated simple data from the function $y = 2x + 8 + \epsilon$ with known noise $\epsilon \sim \mathcal{N}(0, 1)$ (i.e. corrupting the observations with noise with a fixed standard deviation 1), creating datasets increasing in size ranging from 10 data points (example in figure 1e) up to 10,000 data points (example in figure 1f). Knowing the *true* amount of noise in our synthetic dataset, we can assess the quality of the uncertainties predicted by the model.

We used models with three hidden layers of size 1024 and ReLU non-linearities, and repeated each experiment three times, averaging the experiments' results. Figure 1a shows the epistemic uncertainty (in standard deviation) decreasing as the amount of data increases. This uncertainty was computed by generating multiple function draws and evaluating the functions over a test set generated from the same data distribution. Figure 1b shows the aleatoric uncertainty tending towards 1 as the data increases—showing that the model obtains an increasingly improved estimate to the model precision as more data is given. Finally, figure 1c shows the predictive uncertainty obtained by combining the variances of both plots above. This uncertainty seems to converge to a constant value as the epistemic uncertainty decreases and the estimation of the aleatoric uncertainty improves.

Lastly, the optimised dropout probabilities corresponding to the various dataset sizes are given in figure 1d. As can be seen, the optimal dropout probability in each layer decreases as more data is observed, starting from near 0.5 probabilities in all layers with the smallest dataset, and converging to values ranging between 0.2 and 0.4 when 10,000 data points are given to the model. More interesting, the optimal dropout probability for the input layer is constant at near-zero, which is often observed with hand-tuned dropout probabilities as well.

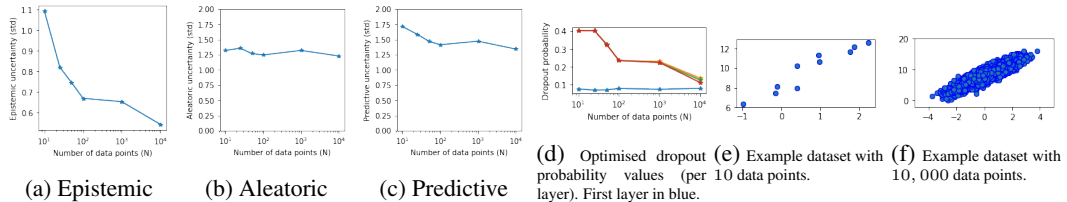

(a) Epistemic     (b) Aleatoric     (c) Predictive     (d) Optimised dropout probability values (per layer). First layer in blue.     (e) Example dataset with 10 data points.     (f) Example dataset with 10,000 data points.

Figure 1: Different uncertainties (epistemic, aleatoric, and predictive, in std) as the number of data points increases, as well as optimised dropout probabilities and example synthetic datasets.

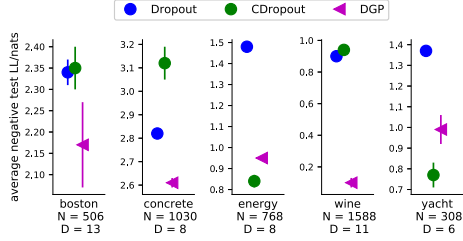

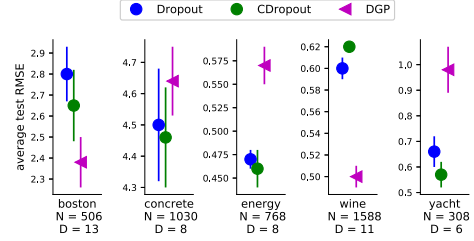

Figure 2: Test negative log likelihood. The lower the better. Best viewed in colour.

Figure 3: Test RMSE. The lower the better. Best viewed in colour.

## 4.2 UCI

We next assess the performance of our technique in a regression setting using the popular UCI benchmark [26]. All experiments were performed using a fully connected neural network (NN) with 2 hidden layers, 50 units each, following the experiment setup of [13]. We compare against a two layer Bayesian NN approximated by standard dropout [9] and a Deep Gaussian Process of depth 2 [4]. Test negative log likelihood for 4 datasets is reported in figure 2, with test error reported in figure 3. Full results as well as experiment setup are given in the appendix D.

Figure 4 shows posterior dropout probabilities across different cross validation splits. Intriguingly, the input layer's dropout probability ($p$) always decreases to essentially zero. This is a recurring pattern we observed with all UCI datasets experiments, and is further discussed in the next section.

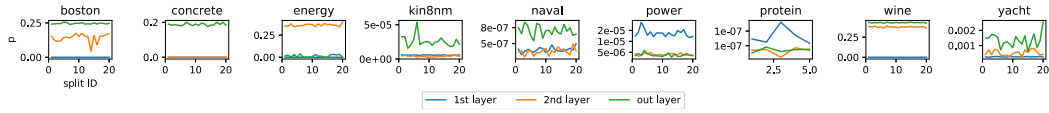

Figure 4: Converged dropout probabilities per layer, split and UCI dataset (best viewed on a computer screen).

## 4.3 MNIST

We further experimented with the standard classification benchmark MNIST [24]. Here we assess the accuracy of Concrete dropout, and study its behaviour in relation to the training set size and *model size*. We assessed a fully connected NN with 3 hidden layers and ReLU activations. All models were trained for 500 epochs ($\sim 2 \cdot 10^5$ iterations); each experiment was run three times using random initial settings in order to avoid reporting spurious results. Concrete dropout achieves MNIST accuracy of $98.6\%$, matching that of hand-tuned dropout.

Figure 5 shows a decrease in converged dropout probabilities as the size of data increases. Notice that while the dropout probabilities in the third hidden and output layers vary by a relatively small amount, they converge to zero in the first two layers. This happens despite the fact that the 2nd and

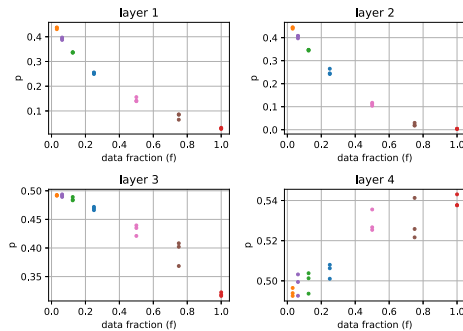

Figure 5: Converged dropout probabilities as function of training set size (3x512 MLP).

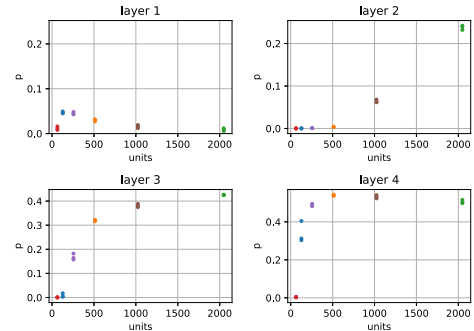

Figure 6: Converged dropout probabilities as function of number of hidden units.

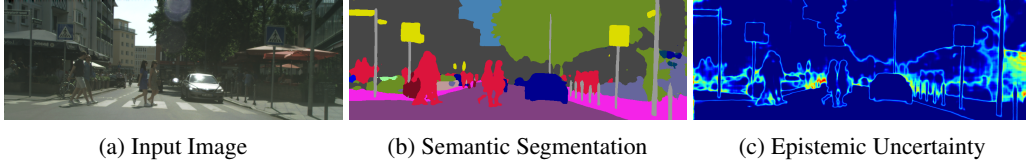

| (a) Input Image | (b) Semantic Segmentation | (c) Epistemic Uncertainty |

Figure 7: Example output from our **semantic segmentation model** (a large computer vision model).

3rd hidden layers are of the same shape and prior length scale setting. Note how the optimal dropout probabilities are zero in the first layer, matching the previous results. However, observe that the model only becomes confident about the optimal input transformation (dropout probabilities are set to zero) after seeing a relatively large number of examples in comparison to the model size (explaining the results in §4.1 where the dropout probabilities of the first layer did not collapse to zero). This implies that removing dropout a priori might lead to suboptimal results if the training set is not sufficiently informative, and it is best to allow the probability to adapt to the data.

Figure 6 provides further insights by comparing the above examined 3x512 MLP model (orange) to other architectures. As can be seen, the dropout probabilities in the first layer stay close to zero, but others steadily increase with the model size as the epistemic uncertainty increases. Further results are given in the appendix D.1.

## 4.4 Computer vision

In computer vision, dropout is typically applied to the final dense layers as a regulariser, because the top layers of the model contain the majority of the model's parameters [32]. For encoder-decoder semantic segmentation models, such as Bayesian SegNet, [21] found through grid-search that the best performing model used dropout over the middle layers (central encoder and decoder units) as they contain the most parameters. However, the vast majority of computer vision models leave the dropout probability fixed at $p = 0.5$, because it is prohibitively expensive to optimise manually – with a few notable exceptions which required considerable computing resources [15, 33].

We demonstrate Concrete dropout's efficacy by applying it to the DenseNet model [17] for semantic segmentation (example input, output, and uncertainty map is given in Figure 7). We use the same training scheme and hyper-parameters as the original authors [17]. We use Concrete dropout weight regulariser $10^{-8}$ (derived from the prior length-scale) and dropout regulariser $0.01 \times N \times H \times W$, where N is the training dataset size, and $H \times W$ are the number of pixels in the image. This is because the loss is pixel-wise, with the random image crops used as model input. The original model uses a hand-tuned dropout $p = 0.2$. Table 1 shows that replacing dropout with Concrete dropout marginally improves performance.

| DenseNet Model Variant | MC Sampling | IoU |
|---|---|---|
| No Dropout | - | 65.8 |
| Dropout (manually-tuned $p = 0.2$) | ✗ | 67.1 |
| Dropout (manually-tuned $p = 0.2$) | ✓ | 67.2 |
| Concrete Dropout | ✗ | 67.2 |
| Concrete Dropout | ✓ | **67.4** |

Table 1: Comparing the performance of Concrete dropout against baseline models with DenseNet [17] on the CamVid road scene semantic segmentation dataset.

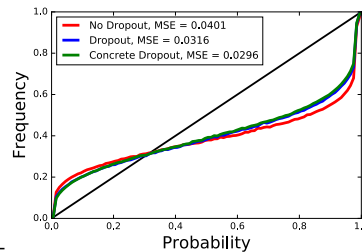

Table 2: Calibration plot. Concrete dropout reduces the uncertainty calibration RMSE compared to the baselines.

**Concrete dropout is tolerant to initialisation values.** Figure 8 shows that for a range of initialisation choices in $p = [0.05, 0.5]$ we converge to a similar optima. Interestingly, we observe that Concrete dropout learns a different pattern to manual dropout tuning results [21]. The second and last layers have larger dropout probability, while the first and middle layers are largely deterministic.

**Concrete dropout improves calibration** of uncertainty obtained from the models. Figure 2 shows calibration plots of a Concrete dropout model against the baselines. This compares the model's predicted uncertainty against the accuracy frequencies, where a perfectly calibrated model corresponds to the line $y = x$.

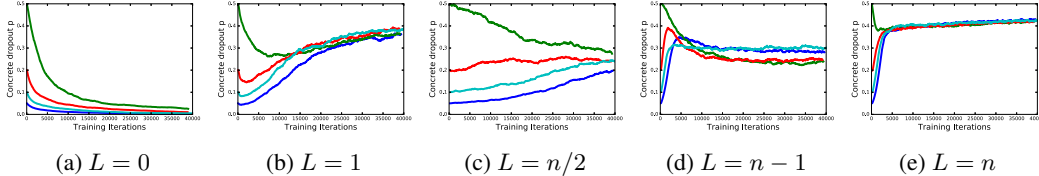

(a) $L = 0$   (b) $L = 1$   (c) $L = n/2$   (d) $L = n - 1$   (e) $L = n$

Figure 8: Learned Concrete dropout probabilities for the first, second, middle and last two layers in a semantic segmentation model. $p$ converges to the same minima for a range of initialisations from $p = [0.05, 0.5]$.

**Concrete dropout layer requires negligible additional compute** compared with standard dropout layers with our implementation. However, using conventional dropout requires considerable resources to manually tune dropout probabilities. Typically, computer vision models consist of $10M+$ parameters, and take multiple days to train on a modern GPU. Using Concrete dropout can decrease the time of model training by weeks by automatically learning the dropout probabilities.

### 4.5 Model-based reinforcement learning

Existing RL research using dropout uncertainty would hold the dropout probability fixed, or decrease it following a schedule [9, 10, 18]. This gives a proxy to the epistemic uncertainty, but raises other difficulties such as planning the dropout schedule. This can also lead to under-exploitation of the environment as was reported in [9] with Thompson sampling. To avoid this under-exploitation, Gal et al. [10] for example performed a grid-search to find $p$ that trades-off this exploration and exploitation over the acquisition of *multiple episodes* at once.

We repeated the experiment setup of [10], where an agent attempts to balance a pendulum hanging from a cart by applying force to the cart. [10] used a fixed dropout probability of $0.1$ in the *dynamics* model. Instead, we use Concrete dropout with the dynamics model, and able to match their cumulative reward (16.5 with 25 time steps). Concrete dropout allows the dropout probability to adapt as more data is collected, instead of being set once and held fixed. Figures 9a–9c show the optimised dropout probabilities per layer vs. the number of episodes (acquired data), as well as the fixed probabilities in the original setup. Concrete dropout automatically decreases the dropout probability as more data is observed. Figures 9d–9g show the dynamics' model epistemic uncertainty for each one of the four state components in the system: $[x, \dot{x}, \theta, \dot{\theta}]$ (cart location, velocity, pendulum angle, and angular velocity). This uncertainty was calculated on a validation set split from the total data after each episode. Note how with Concrete dropout the epistemic uncertainty decreases over time as more data is observed.

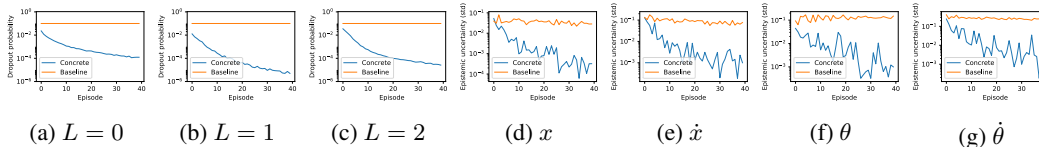

(a) $L = 0$   (b) $L = 1$   (c) $L = 2$   (d) $x$   (e) $\dot{x}$   (f) $\theta$   (g) $\dot{\theta}$

Figure 9: **Concrete dropout in model-based RL**. Left three plots: dropout probabilities for the 3 layers of the dynamics model as a function of the number of episodes (amount of data) observed by the agent (Concrete dropout in blue, baseline in orange). Right four plots: epistemic uncertainty over the dynamics model output for the four state components: $[x, \dot{x}, \theta, \dot{\theta}]$. Best viewed on a computer screen.

## 5 Conclusions and Insights

In this paper we introduced Concrete dropout—a principled extension of dropout which allows for the dropout probabilities to be tuned. We demonstrated improved calibration and uncertainty estimates, as well as reduced experimentation cycle time. Two interesting insights arise from this work. First, common practice in the field where a small dropout probability is often used with the shallow layers of a model seems to be supported by dropout's variational interpretation. This can be seen as evidence towards the variational explanation of dropout. Secondly, an open question arising from previous research was whether dropout works well *because* it forces the weights to be near zero with fixed $p$. Here we showed that allowing $p$ to adapt, gives comparable performance as optimal fixed $p$. Allowing $p$ to change does not force the weight magnitude to be near zero, suggesting that the hypothesis that dropout works *because* $p$ is fixed is false.

## Footnotes

[1]This raises an interesting hypothesis: does dropout work well *because* it forces the weights to be near zero, i.e. regularising the weights? We will comment on this later.

[2]`https://github.com/yaringal/ConcreteDropout`

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
