[Supplementary Material · appendix.pdf]

# A Concrete distribution with a Bernoulli random variable

(a) Relation between $\mathbf{z} \sim \text{Concrete}(p)$ and $u \sim \text{Uniform}[0, 1]$, given by a sigmoid function.

(b) Derivative of $\mathbf{z}$ w.r.t. the dropout parameter $p$.

Figure 10: Concrete distribution with a two-dimensional discrete distribution (Bernoulli) and temperature 0.1.

# B Choice or prior

We use the discrete quantised Gaussian prior suggested in [7]. With this prior the KL divergence between the variational distribution and the prior (last term of eq. (1)) can be evaluated analytically.

With this prior choice our dropout variant can follow two different interpretations. In the first we consider the Concrete distribution as an approximation to the Bernoulli distribution only in the expected log likelihood term (first term in eq. (1)) in order to get derivatives w.r.t. the parameter $p$.

In a second, more interesting, interpretation we regard the Concrete distribution noise itself as a new stochastic regularisation technique (SRT). In this case the expected log likelihood term is not viewed as an approximation, but instead the KL divergence between the variational distribution and the prior is seen as being approximated now. This is because under this view we discretise the Concrete distribution and approximate it as a Bernoulli distribution, in order to evaluate the KL analytically. This holds true as long as the Concrete distribution's temperature is low. However, this temperature parameter can be tuned as well under our variational setting. With larger temperature values the KL approximation would not hold any more. This interesting extension would require us to develop new approximations to the KL for the Concrete distribution, and we leave this as future research.

# C Python code snippet for Concrete Dropout

This Keras wrapper allows learning the dropout probability for any given input layer. Usage:

```
# as the first layer in a model
model = Sequential()
model.add(ConcreteDropout(Dense(8), input_shape=(16)))
# now model.output_shape == (None, 8)
# subsequent layers: no need for input_shape
model.add(ConcreteDropout(Dense(32)))
# now model.output_shape == (None, 32)
```

ConcreteDropout can be used with arbitrary layers, not just Dense, for instance with a Conv2D layer:

```
model = Sequential()
model.add(ConcreteDropout(Conv2D(64, (3, 3)),
input_shape=(299, 299, 3)))
```

although current implementation supports 2D inputs only.

**Arguments**:

- `layer`: a layer instance.
- `weight_regularizer`: A positive number which satisfies

$$\texttt{weight\_regularizer} = l^2/(\tau N)$$

with prior lengthscale $l$, model precision $\tau$ (inverse observation noise), and $N$ the number of instances in the dataset. Note that `kernel_regularizer` is not needed.

- `dropout_regularizer`: A positive number which satisfies

$$\text{dropout\_regularizer} = 2/(\tau N)$$

with model precision $\tau$ (inverse observation noise) and N the number of instances in the dataset.

Note the relation between `dropout_regularizer` and `weight_regularizer`:

$$\text{weight\_regularizer}/\text{dropout\_regularizer} = l^2/2$$

with prior lengthscale $l$. Note also that the factor of two should be ignored for cross-entropy loss, and used only for the Euclidean loss.

Code:

```python
import keras.backend as K
from keras import initializers
from keras.engine import InputSpec
from keras.layers import Dense, Lambda, Wrapper
class ConcreteDropout(Wrapper):
    def __init__(self, layer, weight_regularizer=1e-6,
                 dropout_regularizer=1e-5, **kwargs):
        assert 'kernel_regularizer' not in kwargs
        super(ConcreteDropout, self).__init__(layer, **kwargs)
        self.weight_regularizer = K.cast_to_floatx(weight_regularizer)
        self.dropout_regularizer = K.cast_to_floatx(
            dropout_regularizer)
        self.mc_test_time = mc_test_time
        self.losses = []
        self.supports_masking = True

    def build(self, input_shape=None):
        assert len(input_shape) == 2  # TODO: test with more than two
            dims
        self.input_spec = InputSpec(shape=input_shape)
        if not self.layer.built:
            self.layer.build(input_shape)
            self.layer.built = True
        super(ConcreteDropout, self).build()  # this is very weird..
            we must call super before we add new losses

        # initialise p
        self.p_logit = self.add_weight((1,),
                                       initializers.RandomUniform(-2.,
                                           0.),  # ~0.1 to ~0.5 in
                                           logit space.
                                       name='p_logit',
                                       trainable=True)
        self.p = K.sigmoid(self.p_logit[0])

        # initialise regulariser / prior KL term
        input_dim = input_shape[-1]  # we drop only last dim
        weight = self.layer.kernel
        # Note: we divide by (1 - p) because we scaled layer output by
            (1 - p)
        kernel_regularizer = self.weight_regularizer * K.sum(K.square(
            weight)) / (1. - self.p)
        dropout_regularizer = self.p * K.log(self.p)
        dropout_regularizer += (1. - self.p) * K.log(1. - self.p)
        dropout_regularizer *= self.dropout_regularizer * input_dim
        regularizer = K.sum(kernel_regularizer + dropout_regularizer)
        self.add_loss(regularizer)
```

```python
    def compute_output_shape(self, input_shape):
        return self.layer.compute_output_shape(input_shape)

    def concrete_dropout(self, x):
        eps = K.cast_to_floatx(K.epsilon())
        temp = 1.0 / 10.0
        unif_noise = K.random_uniform(shape=K.shape(x))
        drop_prob = (
            K.log(self.p + eps)
            - K.log(1. - self.p + eps)
            + K.log(unif_noise + eps)
            - K.log(1. - unif_noise + eps)
        )
        drop_prob = K.sigmoid(drop_prob / temp)
        random_tensor = 1. - drop_prob

        retain_prob = 1. - self.p
        x *= random_tensor
        x /= retain_prob
        return x

    def call(self, inputs, training=None):
        return self.layer.call(self.concrete_dropout(inputs))
```

## D   More Results

In the UCI experiments we used length scale $l \cdot \sqrt{K_{\text{in}}}$ where $K_{\text{in}}$ is the number of rows of a particular weight matrix; the constant $l$ was determined manually for each dataset using validation set performance. The reported accuracy and predictive log likelihood were approximated by MC integration using $10^4$ samples, whilst the training time gradients were estimated using a single sample. The training time varied from $10^3$ to $10^4$ to ensure convergence.

The output precision parameter $\tau$ was determined using a variational (MAP-)EM algorithm [3]: the variational parameters (variational E–step), and $\tau$ (M–step) are optimised iteratively by gradient ascent on $(\text{ELBO} + \log p(\tau))$. The $\log p(\tau)$ term only affects the M–step where it replaces the standard MLE by MAP estimation of $\tau$. The prior distribution was set to $p(\tau) = \text{Gamma}(0.1, 0.01)$ for all datasets. Because the optimisation was taking a very long time, presumably due to the high variance of our gradient estimator, we were only using partial E and M–steps (i.e. optimising for only a fixed number of iterations). Empirically, we needed to initialise $\tau$ to be low (i.e. high aleatoric uncertainty) to avoid collapsing into a bad local optima; hence we run the final M–step until convergence which yielded values similar to the ones used by [9]. We report results both prior (*CDropout*) and after (*CDropoutM*) the last M–step. We believe that the optimisation issues might be solved by careful choice of $\tau$'s prior, improved initialisation, or by replacing the gradient optimisation of $\tau$ by Bayesian optimisation.

Full results on all UCI datasets are given next.

| Dataset | N | D | Dropout | CDropout | CDropoutM | DGP |
|---|---|---|---|---|---|---|
| boston | 506 | 13 | -2.34±0.03 | -2.72±0.01 | -2.35±0.05 | ***-2.17±0.10*** |
| concrete | 1030 | 8 | -2.82±0.02 | -3.51±0.00 | -3.12±0.07 | ***-2.61±0.02*** |
| energy | 768 | 8 | -1.48±0.00 | -2.30±0.00 | ***-0.84±0.03*** | -0.95±0.01 |
| kin8nm | 8192 | 8 | 1.10±0.00 | 0.65±0.00 | 1.24±0.00 | ***1.79±0.02*** |
| power | 9568 | 4 | -2.67±0.01 | -2.75±0.01 | -2.75±0.01 | ***-2.58±0.01*** |
| protein | 45730 | 9 | -2.70±0.00 | -2.81±0.00 | -2.81±0.00 | ***-2.11±0.04*** |
| red wine | 1588 | 11 | -0.90±0.01 | -1.70±0.00 | -0.94±0.02 | ***-0.10±0.03*** |
| yacht | 308 | 6 | -1.37±0.02 | -1.75±0.00 | ***-0.77±0.06*** | -0.99±0.07 |
| naval | 11934 | 16 | 4.32±0.00 | 5.87±0.05 | ***5.89±0.05*** | 4.77±0.32 |
| **Average Rank** | | | 2.56±0.23 | 3.56±0.23 | 2.44±0.39 | **1.44±0.23** |

Table 3: Regression experiment: Average test log likelihood/nats

| Dataset | N | D | Dropout | CDropout | CDropoutM | DGP |
|---|---|---|---|---|---|---|
| boston | 506 | 13 | 2.80±0.13 | 2.65±0.17 | 2.65±0.17 | ***2.38±0.12*** |
| concrete | 1030 | 8 | 4.50±0.18 | ***4.46±0.16*** | 4.46±0.16 | 4.64±0.11 |
| energy | 768 | 8 | 0.47±0.01 | ***0.46±0.02*** | 0.46±0.02 | 0.57±0.02 |
| kin8nm | 8192 | 8 | 0.08±0.00 | 0.07±0.00 | 0.07±0.00 | ***0.05±0.00*** |
| power | 9568 | 4 | 3.63±0.04 | 3.70±0.04 | 3.70±0.04 | ***3.60±0.03*** |
| protein | 45730 | 9 | 3.62±0.01 | 3.85±0.02 | 3.85±0.02 | ***3.24±0.10*** |
| red wine | 1588 | 11 | 0.60±0.01 | 0.62±0.00 | 0.62±0.00 | ***0.50±0.01*** |
| yacht | 308 | 6 | 0.66±0.06 | ***0.57±0.05*** | 0.57±0.05 | 0.98±0.09 |
| naval | 11934 | 16 | ***0.00±0.00*** | 0.00±0.00 | 0.00±0.00 | 0.01±0.00 |
| **Average Rank** | | | 2.67±0.31 | **2.00±0.27** | 3.00±0.27 | 2.33±0.50 |

Table 4: Regression experiment: Average test RMSE

## D.1 MNIST

In the MNIST experiments, the data was split into training, validation and testing sets with $5 \cdot 10^4$, $10^4$ and $10^4$ observations respectively.

We used the $l \cdot \sqrt{K_{\mathrm{in}}}$ formula for prior length scale as in section 4.2. The constant $l$ was determined by grid search over $\{10^{-4}, , 10^{-3}, \ldots, 10^0, 2.0\}$; $l = 10^{-2}$ attained the best balance between predictive log likelihood and accuracy. The MC integration schedule was similar to section 4.2 except for using only 200 samples at test time.

Figure 11: Predictive log likelihood and accuracy on test set for different settings of length scale (3x512 MLP).

The right hand side plot in figure 11 shows that our model attains same results as standard dropout. ELBO is a good indicator of optimal length scale if we want to pick a model with best predictive log likelihood. However, we have not observed this correlation for other hyperparameters which concurs with results reported by [7].