[Reviews · NeurIPS 2017]

Reviewer 1



The author formulates dropout as a variational bayes approach (as is also done in a few recent papers). This provides a theoretically grounded way of optimizing the amount of noise per layers. Previous work uses gaussian noise to be able to use reparametrization trick. In this work, a smooth relaxation of the categorical distribution, known as concrete variable or gumbel softmax is used to reduce variance of the gradient. The final algorithm is easy to implement and can be added to any existing architecture at almost no cost. The experimental results shed insights on the amount of noise required with respect to 1) the number of samples 2) the size of each layers 3) the position of the layer in the network. Overall, the paper is well written except for section 3, the most important part. 1) The prior is not specified in the paper and referred to the appendix. In the appendix, there is no explicit equation of the prior. 2) Please use a less compact formulation for the posterior (instead of diag and Bernoulli^{K_l}). It could be rewritten slightly differently and be more obvious to many reader how it relates to conventional dropout. 3) in equation 3, $l$ is undefined. I'm assuming it has nothing to do with the index of the layer. Also, since the prior is not really defined, it's hard to verify this equation. Please specify the prior explicitly and write more steps on how to obtain this equation. 4) Equation 5: I thought that gumbel noise was -log(-log(u)), not +log(u). I might have misunderstood something somewhere. If this is indeed the wrong equation for gumbel noise, please rerun experiments (it should not change much the end-result). 5) The lack of comparison with [23] is disappointing and the justification for doing so is not convincing. 6) I would have loved to see experimentations with a per-weight noise parameter as is done in [23]. Can the author explain why this experiment is not in the paper? ==== post discussion === Overall the paper is good. Please spend the time to make the paper more clear about the issues discussed by the reviewers.

Reviewer 2



This paper attempts to incorporate the dropout rate of Bayesian neural network dropout into the standard gradient based optimization routine. The focus is in relating the dropout rate, p, with a Bernoulli likelihood. To achieve the optimization of p, this is relaxed into the continuous domain through the Concrete distribution, which in this case has a simple sigmoidal form. Experiments test the method in terms of uncertainty modeling and boost in predictive accuracy. In general this is a nicely written paper, but the presentation can be improved further. I did enjoy the intuitive parts, but perhaps they should be a little reduced to make space for some more technical details. For example, the prior p(\omega) in Section 3 and expanding on the form and derivation of eq. (5). I also think it would be better to only refer to "epistemic", "aleatoric" uncertainty in Section 2 and onwards (eg. Fig 1) switch to referring to the proper quantity measured. Technically, this seems like a well motivated work, although I it seems that relaxing the discrete p into (5) moves the approach closer to something like [23]. I was hoping to see a comparison and I do not understand why it has been omitted, even if it is known to underperform. As far as novelty is concerned, this work builds heavily on work by Gal et al. and (inference-wise) moves towards the direction of [23], to some extent. Having said that, there are a few important novel elements, in particular the ability to optimize p as well as some intuitive justifications. The experiments are quite extensive and the code is given at submission time, which is great. In general, the new method doesn't seem to offer a significant boost, but by being robust means that it can speed up experiment time. It would be nice to see some more quantitative proof that the uncertainty is better calibrated, e.g. in active learning (the RL experiment was nice). Finally, the paper would be very much improved if the authors could compare the overall approach to that of tuning the dropout rate with Bayesian Optimization (which is a very common strategy). Some smaller comments: - in line 107- ..., this is implying that w and p have to co-adapt during optimization? - eq. (3): please state clearly what is K - Plots 1b, 1c actually look very similar to me, as opposed to what is stated that one has a trend and the other is flat. - figure 1d is very interesting. - Fig 2: what happens when the error-bar collapses? Why does that happen? Also, how is the standard "Dropout" method tuning p? - lines 250-...: From this argumentation I can still not see why the probabilities collapse to zero. EDIT after rebuttal: ==== I have read the reply by the authors. I believe this will be an interesting read, especially once the comments raised by the reviewers (concerning clarity etc) are addressed.

Reviewer 3



This paper proposes an approach for automatic tuning of dropout probabilities under the variational interpretation of dropout as a Bayesian approximation. Since the dropout probability characterizes the overall posterior uncertainty, such tuning is necessary for calibrated inferences, but has thus far required expensive manual tuning, e.g., via offline grid search. The proposed method replaces the fixed Bernoulli variational distribution at training time with a concrete/gumbel-softmax distribution, allowing use of the reparameterization trick to compute low-variance stochastic gradients of the ELBO wrt dropout probabilities. This is evaluated on a range of tasks ranging from toy synthetic datasets to capturing uncertainty in semantic segmentation and model-based RL. The motivation seems clear and the proposed solution is simple and natural, combining dropout Bayesian inference with the concrete distribution in the obvious way. The technical contributions of the paper seem relatively minimal, but this is not necessarily a problem; it's great when a simple, straightforward method turns out to give impressive results. That said, given the straightforwardness of the approach, the significance of this paper rests heavily on the evaluation, and here I am not really convinced. Although there are a lot of experiments, none seems to show a compelling advantage and collectively they fail to answer many natural questions about the approach. For example: - what is the training time overhead versus standard dropout? (line 281 says 'negligible' but this is never quantified, naively I'd expect differentiating through the gumbel-softmax to add significant overhead) - do the learned dropout probabilities match the (presumably optimal) probabilities obtained via grid search? - how to set the temperature of the concrete distribution (and does this matter)? The code in the appendix seems to used a fixed t=0.1 but this is never mentioned or discussed in the paper. - how does this method compare to other reasonable baselines, eg explicit mean-field BNNs, or Gaussian dropout with tuned alpha? The latter seems like a highly relevant comparison, since the Gaussian approach seems simpler both theoretically and in implementation (is directly reparameterizable without the concrete approximation), and to dismiss it by reference to a brief workshop paper [28] claiming the approach 'underperforms' (I don't see such a conclusion in that paper even for the tasks it considers, let alone in any more general sense) is lazy. - there is a fair amount of sloppiness in the writing, including multiple typos and cases where text and figures don't agree (some detailed below). This does not build confidence in the meticulousness of the experiments. More specific comments: Synthetic data (sec 4.1): I am not sure what to conclude from this experiment. It reads as simply presenting the behavior of the proposed approach, without any critical evaluation of whether that behavior is good. Eg, Fig 1b describes aleatoric uncertainty as following an 'increasing trend' (a bit of a stretch given that the majority of steps are decreasing), and 1c describes predictive uncertainty as 'mostly constant' (though the overall shape actually looks very similar to 1b) -- but why is it obvious that either of these behaviors is correct or desirable? Even if this behavior qualitatively matches what you'd get from a more principled Bayesian approach (I was expecting the paper to argue this but it doesn't), a simple synthetic task should be an opportunity for a real quantitative comparison: how closely do the dropout uncertainties match what you'd get from, e.g., HMC, or even exact Bayesian linear regression (feasible here since the true function is linear -- so concrete dropout on a linear model should be a fair comparison). UCI (sec 4.2): the experimental setup is not clear to me -- is the 'dropout' baseline using some fixed probability (in which case, what is it?) or is it tuned by grid search (how?). More fundamentally, given that the goal of CDropout is to produce calibrated uncertainties, why are its NLL scores (fig 2, a reasonable measure of uncertainty) *worse* than standard dropout in the majority of cases? MNIST (sec 4.3): again no real comparison to other approaches, no evaluation of uncertainty quality. The accuracy 'matches' hand-tuned dropout -- does it find the same probabilities? If so this is worth saying. If not, seems strange to refer to the converged probabilities as 'optimal' and analyze them for insights. Computer vision (sec 4.4): this is to me the most promising experiment, showing that concrete dropout matches or slightly beats hand-tuned dropout on a real task. It's especially nice to see that the learned probabilities are robust to initialization since this is a natural worry in non-convex optimization. Some things that seem confusing here: - what is the metric "IoU"? presumably Intersection over Union but this is never spelled out or explained. - the differences in calibration between methods are relatively small compared to the overall miscalibration of all the methods - why is this? - text says "middle layers are largely deterministic" which seems inconsistent with Fig 8c showing convergence to around p=0.2? (nitpick: "Table" 2 is actually a figure) Model-based RL (sec 4.5): It's nice that the model matches our intuition that dynamics uncertainty should decrease with experience. But it sounds like this tuning doesn't actually improve performance (only matches the cumulative reward of a fixed 0.1 dropout), so what's the point? Presumably there should be some advantages to better-calibrated dynamics uncertainty; why do they not show up in this task? The method proposed in this paper seems promising and potentially quite useful. If so it should be possible to put together a careful and clear evaluation showing compelling advantages (and minor costs) over relevant baselines. I'd like to read that paper! But as currently submitted I don't think this meets the NIPS bar. UPDATE: increasing my rating slightly post rebuttal. I still think this paper is marginal, but the clarifications promised in the rebuttal (describing the tuning of dropout baselines, and the problems with Gaussian dropout, among others) will make it stronger. It's worth noting that recent work (Molchanov et al., ICML 2017 https://arxiv.org/abs/1701.05369) claims to fix the variance and degeneracy issues with Gaussian dropout; this would be an interesting comparison for future work.